# Comparing Influenza Virus Biology for Understanding Influenza D Virus

**DOI:** 10.3390/v14051036

**Published:** 2022-05-13

**Authors:** Raegan M. Skelton, Victor C. Huber

**Affiliations:** Division of Basic Biomedical Sciences, University of South Dakota, Vermillion, SD 57069, USA; raegan.skelton@coyotes.usd.edu

**Keywords:** influenza D virus, innate immunity, interferon, virus pathogenesis

## Abstract

The newest type of influenza virus, influenza D virus (IDV), was isolated in 2011. IDV circulates in several animal species worldwide, causing mild respiratory illness in its natural hosts. Importantly, IDV does not cause clinical disease in humans and does not spread easily from person to person. Here, we review what is known about the host–pathogen interactions that may limit IDV illness. We focus on early immune interactions between the virus and infected host cells in our summary of what is known about IDV pathogenesis. This work establishes a foundation for future research into IDV infection and immunity in mammalian hosts.

## 1. Introduction

Within the *Orthomyxoviridae* family, the four genera of influenza viruses are *Alphainfluenzavirus*, *Betainfluenzavirus*, *Gammainfluenzavirus*, and *Deltainfluenzavirus*, each consisting of just one species: influenza A, B, C, and D, respectively [1]. The most aggressive and noteworthy of the influenza viruses, influenza A viruses (IAVs) are responsible for seasonal epidemics and all previous influenza pandemics [2]. Of the four types of influenza viruses, IAVs also have the widest host range, highest associated morbidity and mortality, and are capable of causing the most severe disease [1,3]. For these reasons, IAVs are the influenza viruses that have been studied the most. Influenza B viruses (IBVs) can cause seasonal epidemics with mild to severe illness, and are usually limited to disease in humans [4]. Alternatively, while most individuals develop antibodies against influenza C viruses (ICVs) early in life (indicating prior exposure to this virus), there are relatively few reports of clinical disease caused by ICV [4]. Discovered in 2011, the newest members of the *Orthomyxoviridae* family, influenza D viruses (IDV), have not yet been associated with clinical infection in humans [5,6]. Interestingly, humans can become seropositive for IDV, despite the absence of clinical symptoms [6,7,8]. The incidence of seropositivity increases in individuals that are at high risk for exposure to this virus, including those in close contact with cattle [7]. With little known about the host–pathogen interactions with IDV, this virus presents an exciting opportunity to evaluate host responses against a virus that does not cause illness in humans. By comparing these responses with those that develop against more aggressive IAVs, we can gain unique insight into how the host immune response can effectively fight—or fail to fight—influenza virus infections.

## 2. General Features of Influenza Virus Infections

Influenza is an acute and contagious respiratory disease that can cause mild to severe illness [2,9]. The viruses responsible for these infections are in the *Orthomyxoviridae* family [10], and four genera from this family are of interest: *Alphainfluenzavirus* (influenza A virus, IAV), *Betainfluenzavirus* (influenza B virus, IBV), *Gammainfluenzavirus* (influenza C virus, ICV), and *Deltainfluenzavirus* (influenza D virus, IDV) [2,11]. Within the human host, influenza-like illness has been observed during infection with IAV, IBV, and ICV, but not with IDV [1]. Symptoms of influenza infection can develop suddenly, and mild infections remain confined mostly to the upper respiratory tract with a sore throat, runny nose, and cough as well as fever, headache, muscle pain, and fatigue [2,9,12]. More severe infections, typically associated with either IAV or IBV, can spread to the lower respiratory tract and other areas in the body, leading to more serious complications such as pneumonia, secondary bacterial infections, myocarditis, encephalitis, and even death [2,13]. Although symptoms usually last for three to eight days, a persistent cough and fatigue can remain for at least two weeks [10,12]. Seasonal influenza continues to have a significant impact in the United States, with the Centers for Disease Control and Prevention (CDC) estimating that the 2018–2019 influenza season (prior to the start of the COVID-19 pandemic) had 36 million illnesses, nearly 500,000 hospitalizations, and over 34,000 influenza-associated deaths [9]. Influenza has a far-reaching impact and is also a significant global health concern. Worldwide, there are an estimated 1 billion infections, 54.5 million hospitalizations, and 200,000 influenza-associated deaths every year due to influenza-like illnesses [14,15]. According to the World Health Organization (WHO), about 10% of the world’s population is affected by influenza annually [15].

## 3. Host Range of Influenza Viruses

Depending on the genus, the infection cycle of influenza viruses can take place in a wide variety of host species (Figure 1). IAVs have the largest host range and are considered to be “species jumpers” for their ability to rapidly move between host species [16].

In nature, the IAV host range includes birds, which serve as their natural reservoir, as well as humans, pigs, rodents, cats, horses, and even whales [2,16]. Seasonal and epidemic influenza virus infections can be caused by IAV strains, and all previous influenza pandemics have been due to IAVs [2,17]. In contrast, IBV and ICV infect a narrow host range. IBVs, which can also cause seasonal and epidemic outbreaks of influenza, are mostly restricted to the human host, but they can also infect pigs and seals [2,16]. Humans and pigs are the only known hosts for ICV, and most humans acquire immunity against ICV by early childhood while exhibiting either mild symptoms or asymptomatic illness [16,18,19].

In contrast with the other three influenza virus types, IDV has not yet been described clinically in humans. This virus naturally infects cattle, pigs, horses, and other ruminants around the world such as camels, goats, and sheep [5,6,20,21]. As research focuses on this new influenza virus type, the potential host range of IDV continues to increase. In the research setting, mammalian infection by IDV has been demonstrated in mice [22,23], guinea pigs [24], and ferrets [6], and in vitro studies show that IDV can infect a range of cells while replicating at different temperatures [5,6,25]. In the absence of known clinical infection, seropositivity against IDV remains our only evidence that IDV can infect humans. Studies using archived sera have detected antibodies against IDV at least as far back as 2005, pre-dating the isolation of this virus [8]. In fact, a published report showed that greater than 90% of those who regularly work with cattle have antibodies against IDV [7], suggesting that IDV is capable of inducing adaptive immunity in humans that have high exposure rates [5]. In the absence of clinical cases of IDV, it is assumed that these infections are associated with either mild symptoms or asymptomatic illness. Thus, at this time neither the full host range nor the breadth of disease caused by IDV are completely known, and additional research is needed to fully appreciate the global reach of this newly identified influenza virus type.

## 4. Influenza Virus Proteins and Their Function

All four types of influenza virus are enveloped, take on a spherical (~100 nm) or filamentous (300 nm+) shape, and contain a single-stranded, segmented RNA genome [4,11,26]. While IAV and IBV use eight RNA gene segments to express anywhere from 10–17 proteins, ICV and IDV utilize seven RNA gene segments that express nine proteins [4,27,28]. Influenza viruses are packaged with everything needed to infect a host cell and make more virus, with replication cycles highly conserved across all four types of influenza virus [4]. The first step in the infection cycle is binding and entry, when influenza viruses use surface proteins to bind sialic acids or sialic acid derivatives located on host cell receptors [4,29,30]. For IAV and IBV, the surface protein that targets host cells is hemagglutinin (HA), while hemagglutinin esterase fusion (HEF) is used by ICV and IDV. HA and HEF binding targets can be found within the respiratory tract of mammalian hosts, as well as in the intestinal tract of avian hosts [2]. The interaction between the virus and the host cell triggers receptor-mediated endocytosis of the virus into the host cell and initiation of the infection cycle [4,30].

Upon entry into a host cell, ion channels along the surface of the virus function similarly across influenza virus types (M2 for IAV, BM2 for IBV, CM2 for ICV, and DM2 for IDV), allowing for acidification of the endosome to facilitate fusion of viral and endosomal membranes. Membrane fusion allows for release of viral RNA gene segments, packaged as viral ribonucleoprotein complexes (vRNPs), into the host cell cytoplasm. Each vRNP (Figure 2) consists of an RNA gene segment neatly wrapped with several copies of influenza nucleoprotein (NP) and bound by the viral RNA-dependent RNA polymerase (RDRP) complex, which consists of PB2, PB1, and PA for IAV/IBV or PB2, PB1, and P3 for ICV/IDV [4,30]. The vRNPs are transported into the nucleus where the RDRP complex bound to each RNA segment initiates replication and transcription of viral RNA (vRNA) [30]. The vRNA that enter the nucleus as negative-sense templates are transcribed by the RDRP complex into “host” positive-sense mRNA that are translated into viral proteins within the cytoplasm using the host’s ribosomal machinery [4,30]. Newly formed vRNPs are transported out of the nucleus with help from the influenza nuclear export protein (NEP), expressed by all four influenza virus types, while appropriate viral are transported to the cell surface. These viral proteins work together to package the newly created proteins and genetic material into budding virions [30].

Each RNA gene segment packaged in an influenza virus encodes at least one protein that will help facilitate viral replication, with some gene segments encoding accessory proteins that perform additional functions within infected cells. The names and functions of the major protein products encoded by each influenza gene segment are summarized in Table 1. Accessory proteins of influenza A viruses are reviewed in more detail elsewhere [27,31].

All four influenza virus types use the M1 protein to package nascent virions with vRNPs and the virions then bud from the host cell with everything they need to infect new cells [4,30]. At budding, virus-binding residues on the host cell surface are cleaved through either neuraminidase (NA) activity for IAV and IBV or the esterase function of the HEF protein for ICV and IDV [4,30]. One of the unique features of ICV and IDV is that the single HEF protein performs both the binding and residue-cleaving functions of the IAV and IBV HA and NA proteins [4]. Ultimately, this allows ICV and IDV to utilize only seven RNA segments rather than the eight RNA segments required for IAV and IBV. Interestingly, the mild illness that ICV causes may be partly due to the fact that folding of the ICV HEF protein in infected host cells is more efficient at the lower temperature of 33 °C that is found in the upper respiratory tract. This temperature sensitivity may limit progression of ICV into the warmer lower respiratory tract [4]. It is interesting that, unlike the HEF protein of ICV, the IDV HEF protein demonstrates surprising temperature and acid stability [32]. In fact, the IDV HEF is the most stable surface protein expressed across all influenza virus types and it may contribute to the ability of IDVs to infect and spread between host species [4]. Thus, the HEF protein presents a unique feature of IDV that sets it apart from the other three types of influenza viruses, and its potential etiological significance should continue to be explored.

## 5. Innate Immunity against Influenza

In the human host, IAVs continue to cause significant morbidity and mortality each year, while IDVs have not been associated with clinical illness. Long before antibodies that are highly specific for a given pathogen can be produced, many early signals of the innate immune response serve as the first line of defense against infection [33,34,35]. Activation of this highly specialized attack system with multiple levels of protection, communication, and functionality involves two critical antiviral pathways that are activated within hours of a viral infection: the interferon regulatory factor (IRF) pathway and the nuclear factor kappa-light-chain-enhancer of activated B cells (NFκB) pathway [33,34]. The IRF and NFκB pathways induce cytokine expression from host cells, and these cytokines act as key communicators of the immune response [34,35,36]. The most important agents produced during viral infections are the cytokines known as interferons (IFNs), which induce hundreds of IFN-stimulated genes (ISGs) and effectively initiate the antiviral state within infected tissues. Through induction of ISGs and additional cytokines, IFNs elicit specific functions including: (1) telling neighboring infected cells to undergo apoptosis [37,38], (2) altering receptor expression on the surface of infected cells to increase recognition and destruction by surrounding immune cells [37,38], (3) signaling neighboring uninfected cells to destroy RNA and reduce protein synthesis to limit spread of a viral infection [36,38], and (4) increasing the recruitment and activation of innate immune response cells to more effectively eliminate the viral infection [37,39,40]. Activation and secretion of IFN by infected cells can be visualized in Figure 3, using infected lung epithelial cells as an example.

### Influenza Evasion of Host Innate Immunity

Due to the effectiveness of innate immunity through signals such as IFN, influenza viruses have evolved to express several proteins that counteract these host responses. Understanding host–pathogen interactions against IDV is critical for understanding why pathogenesis differs between IDV and other types of influenza virus. Protein components of the RNA-dependent RNA polymerase of both IAV and IBV, as well as the M1 protein of IAV, have all demonstrated antagonism of IFN response [4,11,42,43,44,45,46]. The IAV accessory proteins PB1-F2 and PA-X each stem from an alternate reading frame or single frameshift of the polymerase protein components of IAVs and have also been linked to anti-immune functions. PB1-F2 is a virulence factor for IAVs [47,48,49,50,51], demonstrating increased cell death [52], reduced viral clearance [51], and enhancement of viral gene expression [48,53,54] in infected hosts. Regarding PA-X, this protein not only decreases type I IFN expression by preventing innate immune receptor signaling [55], it also facilitates degradation of host mRNAs without affecting vRNAs [56,57,58]. The most notable and well-studied innate immune antagonist expressed by influenza viruses is the nonstructural 1 (NS1) protein, a multifunctional NS gene product that utilizes specific domains to antagonize numerous components of the host innate immune response [59]. For example, the NS1 protein of influenza A and B viruses lead to global effects within infected cells that block host gene expression, particularly at the level of the host IFN response while simultaneously promoting expression of viral genes, as reviewed in [27,60].

While accessory proteins such as PB1-F2 and PA-X have yet to be described for IDV, the literature has identified two IDV proteins with anti-immune function: matrix 1 (M1) and nonstructural 1 (NS1). As with other influenza virus types, the M1 protein of IDV is encoded on the matrix (M) gene. Recently, the M1 protein of IDV was found to negatively regulate type I IFN signaling using a similar mechanism to the M1 protein of IAV [46]. So far, the NS1 protein of IDV has only proved similar to its IAV and IBV analogs through its ability to interfere with innate immunity in vitro, while research suggests this immune response antagonism is lost in vivo [41,61]. Nogales, et al. (2019) demonstrated for the first time that IAVs encoding the NS1 protein of IDV could antagonize IFN-β responses in human lung epithelial cells in vitro similarly to wildtype IAV [61]. However, the IDV NS1 protein-encoding IAVs did not demonstrate the same virulence or viral replication in vivo as wildtype IAV infection. As the NS1 protein of IAV is well-known for being multifunctional (capable of interacting with many different factors of host signaling pathways within infected cells), this work suggests functional differences exist between the IAV NS1 protein and NS1 proteins of other influenza virus types such as IDV [61]. Because IDV is the only type of influenza virus that is not known to cause clinical disease in the human host, it is of particular interest to understand host–pathogen interactions that determine virus infectivity as compared to other types of influenza virus. Additional research exploring the possible anti-immune functions of other IDV genes, accessory proteins of IDV, and the relationship between the NS1 protein of IDV and host immune response pathways will allow us to better understand what may be driving IDV’s lack of virulence in human hosts.

## 6. Importance of Studying Influenza D Virus

IDV has not been shown to cause clinical signs or symptoms of disease within the human host and there is not as much known about how the immune system responds to this novel influenza virus type [5]. So far, researchers have made a handful of interesting observations that warrant further investigation about IDV, but many questions remain. For example, we still do not know the full host range of IDV and do not know the breadth of disease caused by IDV in these hosts. At the viral gene level, it remains unclear how the increased stability of the IDV HEF protein impacts its ability to spread and persist in infected hosts, and how interactions between the IDV NS1 protein and the host influence innate immune responses. Since IAV and IBV remain the most aggressive influenza viruses in the human host, there has been a lot of research into the host–pathogen interactions induced by these viruses. Using this information to begin understanding host–pathogen interactions with IDV will allow us to determine what causes the lack of illness associated with IDV in the human host. This review identifies some of the complex interactions that exist between host and virus, with emphasis on the critical role of innate immunity. While there is still much to learn about IDV, we can use our knowledge of IAV, IBV, and ICV to guide future research efforts.

## Figures and Tables

**Figure 1 viruses-14-01036-f001:**
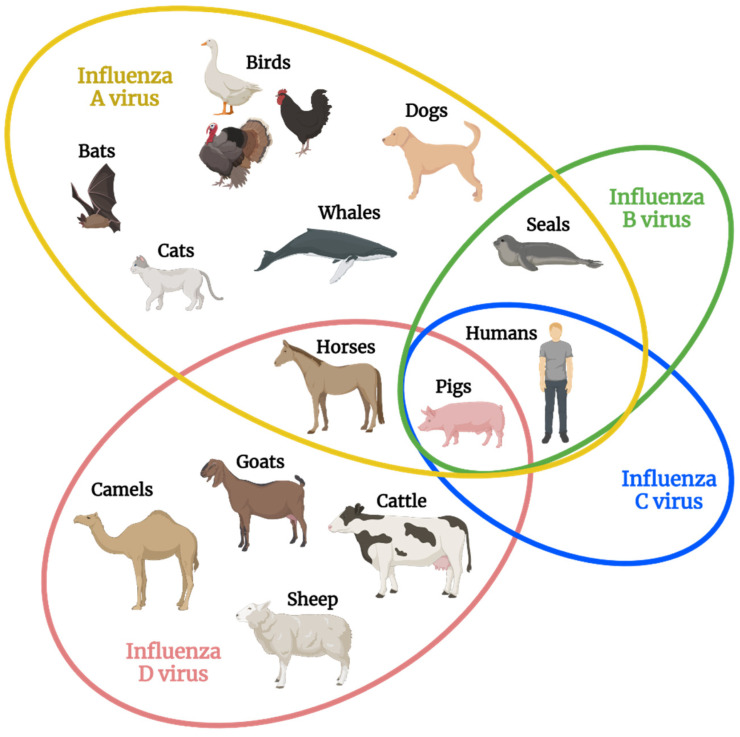
Natural host range of each influenza virus type. The major species of hosts that each influenza virus type naturally infects is shown, with some overlap existing across influenza virus types. Of note, pigs are the only species known to be infected by all four influenza virus types, and influenza D virus demonstrates the second widest host range behind influenza A virus [16]. Figure created with BioRender.com was adapted from ref. [16]: Kuchipudi and Nissly, 2018.

**Figure 2 viruses-14-01036-f002:**
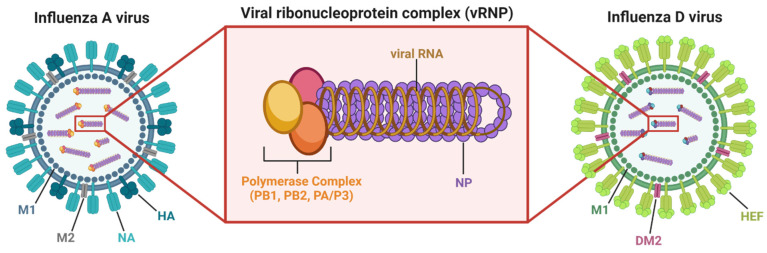
Influenza virus and viral ribonucleoprotein (vRNP) structure. Examples of IAV and IDV are shown, with vRNP structure for IAV magnified. IAVs are studded with the cell surface proteins hemagglutinin (HA) and neuraminidase (NA), while IDVs are studded with the cell surface protein hemagglutinin esterase fusion (HEF). Influenza viruses also have ion channels (M2 for IAV, DM2 for IDV) along their surface and contain an inner matrix of envelope support proteins (M1 for IAV and IDV). As shown for IAV, each of the RNA gene segments are individually wrapped into vRNP complexes which consist of the viral polymerase complex (for IAV, proteins PB1, PB2, and PA) bound to a viral RNA (vRNA) segment neatly surrounded by several copies of viral nucleoprotein (NP). This structure helps initiate transcription and replication of the viral RNA genome within the nucleus of an infected host cell [30]. Figure created with BioRender.com was generated with assistance from ref. [30]: Dou et al., 2018.

**Figure 3 viruses-14-01036-f003:**
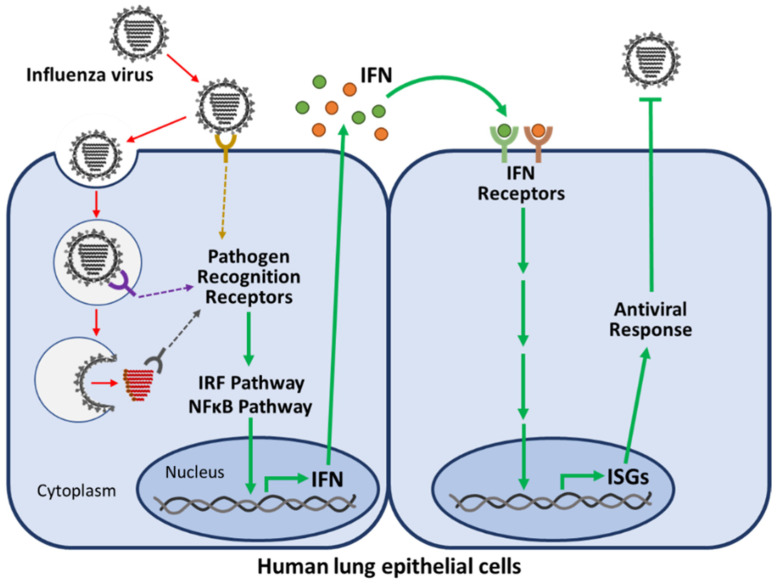
Key players in the innate immune response. Human lung epithelial cells are shown as an example for infected host cells. The innate immune response detects a viral infection at multiple points in the infection cycle through pathogen recognition receptors. These receptors activate the IRF and NFκB pathways, leading to transcription factors that will enter the nucleus and induce the expression of antiviral interferons (IFNs). These IFNs will be secreted from the infected cell and signal through IFN receptors to induce expression of IFN-stimulated genes (ISGs), establishing an antiviral state within the host to limit virus replication and spread [39,41]. (Author dissertation. Labels “nucleus” and “cytoplasm” added from original Dissertation Figure 1.4, ProQuest Document ID 2572559761).

**Table 1 viruses-14-01036-t001:** Influenza virus genes and proteins. Gene segments are listed for influenza A virus (IAV), influenza B virus (IBV), influenza C virus (ICV), and influenza D virus (IDV). The protein product(s) of each segment and their associated functions are summarized for each influenza species [4,27,31].

Gene Segment	IAV	IBV	Viral Function
1	PB2	PB2	RNA-dependent RNA polymerase (RDRP) component
2	PB1	PB1	RDRP component
	PB1-F2 ^1^		Inflammation, apoptosis, regulation of host immune responses
	PB1-N40 ^1^		Regulates PB1 expression and activity
3	PA	PA	RDRP component
	PA-X ^1^		Enhances viral gene expression, facilitates host mRNA degradation, regulation of cell-mediated host responses
	PA-N155 ^1^		Functions unknown, likely involved with viral replication
	PA-N182 ^1^		Functions unknown, likely involved with viral replication
4	HA	HA	Host receptor binding and membrane fusion
5	NP	NP	Packages viral RNA in vRNPs ^2^ with RDRP components
6	NA	NA	Sialidase; assists with release of new virions from host cell
		NB	Function unknown but highly conserved
7	M1	M1	Facilitates packing of vRNPs into new virions
	M2	BM2	Ion channel; assists in release of vRNPs into host cytoplasm
	M42 ^1^		Alternate ion channel
8	NS1	NS1	Host immune response antagonism
	NS2/NEP	NS2/NEP	Nuclear export protein for newly synthesized vRNPs
**Gene Segment**	**ICV**	**IDV**	**Viral Function**
1	PB2	PB2	RNA-dependent RNA polymerase (RDRP) component
2	PB1	PB1	RDRP component
3	P3	P3	RDRP component
4	HEF	HEF	Host receptor binding, membrane fusion; esterase; assists release of new virions
5	NP	NP	Packages viral RNA in vRNPs with RDRP components
6	M1	M1	Facilitates packing of vRNPs into new virions
	CM2	DM2	Ion channel; assists in release of vRNPs into host cytoplasm
7	NS1	NS1	Host immune response antagonism
	NS2	NS2	Nuclear export protein for newly synthesized vRNPs

^1^ Accessory protein; ^2^ vRNPs: viral ribonucleoproteins.

## Data Availability

Not applicable.

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
