# Peer review of "Comparing Influenza Virus Biology for Understanding Influenza D Virus"

_viruses, 2022, doi:10.3390/v14051036_

Round 1

Reviewer 1 Report

This manuscript describes host immune responses revealed in immunological research and the proteins of Influenza viruses that can modify the host immune responses. It is thought interesting and worth to see viral infection from both immunological and virological aspects, however The description of known functions of viral protein from IDV and the rationale to extrapolate those findings to IDV are limited. Moreover, this manuscript contained too much general information of influenza viruses and innate immunity and few insights to understand “host:IDV interactions”. These are unbalanced and redundant to obtain the information that is considered necessary. Therefore, the manuscript appears suitable more general journal rather than virological specific one.

Major comments

  1. Title. Modify to one suiting the contents as the manuscript is focusing on only immunological interaction between hosts and Influenza viruses.

Minor comments

  1. Page 3, line 6 from the bottom. “Negative-sense vRNA” is correct?

  1. Page 6 line from the bottom. Spell out when you use abbreviation (IFNAR1/IFNAR2 in this case).

  1. Lines 495-498 and 542-545, The description of what they are currently working on does not belong in this article and should be removed.

Author Response

Letter to Reviewers

The authors thank both reviewers for the positive review of our manuscript. The feedback provided has improved the presentation of the data. We have addressed all Reviewers’ comments both in the manuscript and below in italics.

Reviewers' Comments to the Authors:

REVIEWER 1

This manuscript describes host immune responses revealed in immunological research and the proteins of Influenza viruses that can modify the host immune responses. It is thought interesting and worth to see viral infection from both immunological and virological aspects, however The description of known functions of viral protein from IDV and the rationale to extrapolate those findings to IDV are limited. Moreover, this manuscript contained too much general information of influenza viruses and innate immunity and few insights to understand “host:IDV interactions”. These are unbalanced and redundant to obtain the information that is considered necessary. Therefore, the manuscript appears suitable more general journal rather than virological specific one.

Author Response: We thank the reviewer for this comment, and agree that substantial revision was required to both restructure and streamline the manuscript to develop a more focused review.  We have made substantial revisions to the manuscript, and now feel that the comparisons between IAV, IBV, ICV, and IDV are more clearly detailed.  In particular, we have focused more specifically on the immune responses against these viruses with emphasis on innate immunity.  In addition, we have limited the general information regarding influenza viruses to focus more on innate host:IDV interactions.

Major comments 

  1. Title. Modify to one suiting the contents as the manuscript is focusing on only immunological interaction between hosts and Influenza viruses. 

Author Response: We have revised the title to Reviewing Potential Contributions of Host:Pathogen Interactions Toward Influenza D Virus Infection and Immunity”, which we feel is a title that more appropriately captures the revised focus of our review.

Minor comments 

  1. Page 3, line 6 from the bottom. “Negative-sense vRNA” is correct? 

Author Response: Thank you for pointing this out.  We have edited the text to read: “The vRNA that enter the nucleus as negative-sense templates are transcribed by the RDRP complex into “host” positive-sense mRNA that will be translated into viral proteins within the cytoplasm, using host ribosomal machinery.”

  1. Page 6 line from the bottom. Spell out when you use abbreviation (IFNAR1/IFNAR2 in this case). 

Author response: Thank you for bringing this to our attention.  We have spelled out these abbreviations on first use in the text.

  1. Lines 495-498 and 542-545, The description of what they are currently working on does not belong in this article and should be removed.

Author response: We have removed descriptions related to our current research from the revised manuscript.

Reviewer 2 Report

In this manuscript, authors aimed to present current understanding of influenza A virus and its interaction with hosts and relate that with the little known influenza D virus. The authors have described in detail about influenza viruses, their replication, and host immune responses. However, to fulfill the aim of the manuscript, following changes are requested:

  1. Restructure the manuscript to reflect the objective: authors need to present this manuscript as a comparative study. Provide insights about the host range, replication pattern, and host responses related to influenza A viruses and within each section provide the current understanding of similar with regards to the influenza D virus. By doing this, authors can present the existing gaps and need of further experiments in influenza D virus research field. 
  2. Prepare figures of your own: the researchers have copied almost all figures from other articles. I strongly recommend the authors to create figures of their own, reflecting their perspectives based on the evidences, using softwares like Biorender. I also suggest to present a figure related to influenza D virus where known and unknowns are highlighted in terms of virus replication and host responses. 

Minor: Lines 495 - 498, if there are no relevant citations, it would be better to remove claims like 'Our study is currently investigating'. Rather, this can be used as a gap in research field. 

Author Response

Letter to Reviewers

The authors thank both reviewers for the positive review of our manuscript. The feedback provided has improved the presentation of the data. We have addressed all Reviewers’ comments both in the manuscript and below in italics.

Reviewers' Comments to the Authors:

REVIEWER 2

In this manuscript, authors aimed to present current understanding of influenza A virus and its interaction with hosts and relate that with the little known influenza D virus. The authors have described in detail about influenza viruses, their replication, and host immune responses. However, to fulfill the aim of the manuscript, following changes are requested:

  1. Restructure the manuscript to reflect the objective: authors need to present this manuscript as a comparative study. Provide insights about the host range, replication pattern, and host responses related to influenza A viruses and within each section provide the current understanding of similar with regards to the influenza D virus. By doing this, authors can present the existing gaps and need of further experiments in influenza D virus research field. 

Author response: Thank you for the thoughtful comments for improving the manuscript.  As described in our response to Reviewer 1, we have substantially restructured and streamlined the manuscript to tell a more focused story that compares what is know about IAV with what is known about IDV.  Our efforts have hopefully achieved our goal of identifying gaps in our knowledge of IDV that will help future research efforts toward understanding the pathogenesis of IDV.

  1. Prepare figures of your own: the researchers have copied almost all figures from other articles. I strongly recommend the authors to create figures of their own, reflecting their perspectives based on the evidences, using softwares like Biorender. I also suggest to present a figure related to influenza D virus where known and unknowns are highlighted in terms of virus replication and host responses. 

Author response: We have edited our figures as requested.  This has helped us create figures that more closely align with the manuscript and the information presented in its current form.  Thank you for this recommendation.

Minor: Lines 495 - 498, if there are no relevant citations, it would be better to remove claims like 'Our study is currently investigating'. Rather, this can be used as a gap in research field.

Author response: We have removed descriptions related to our current research from the revised manuscript.

Round 2

Reviewer 1 Report

I felt that this review article focused primarily on NS1 function as an interferon antagonist for influenza A viruses rather than those for influenza D virus. It is difficult to make this a review article because there is still very little research on NS1 function for D influenza viruses. Thus, the content of this review is not the subject matter that its title indicates. There have been many review articles on NS1 of influenza A viruses, and we believe that there is little need to publish this review article again in this journal.

Author Response

REVIEWER 1

I felt that this review article focused primarily on NS1 function as an interferon antagonist for influenza A viruses rather than those for influenza D virus. It is difficult to make this a review article because there is still very little research on NS1 function for D influenza viruses. Thus, the content of this review is not the subject matter that its title indicates. There have been many review articles on NS1 of influenza A viruses, and we believe that there is little need to publish this review article again in this journal.

Author Response: We thank the reviewer for this additional comment.  We agree with this comment, and further revised the manuscript by shortening the entire "Evasion of host immune responses” section to focus on what is known about IDV gene interactions with the host. In addition, we reviewed the section on IFN and thought that also reviewed information that was out of context in a review that focuses on IDV.  We deleted most of the section describing the history and identification of the different types of IFN, and this allows the information to focus more directly on potential IDV interactions within an infected host. 

Reviewer 2 Report

The earlier comments raised are addressed and the manuscript looks better organized compared to earlier version. For figure 1, the authors changed the figure but it does not agree with the text and facts. E.g., as the authors mentioned humans are not infected with influenza D but stil they have in figure 1, included humans within the influenza D circle. Same way other animals are also included which are not naturally infected. These need to be reconfirmed and addressed. 

Author Response

REVIEWER 2

The earlier comments raised are addressed and the manuscript looks better organized compared to earlier version. For figure 1, the authors changed the figure but it does not agree with the text and facts. E.g., as the authors mentioned humans are not infected with influenza D but stil they have in figure 1, included humans within the influenza D circle. Same way other animals are also included which are not naturally infected. These need to be reconfirmed and addressed. 

Author response: Thank you for the positive feedback.  We have now edited Figure 1 and the text describing it to focus on the “naturally-infected” hosts for different influenza virus types.  We also confirmed that all the hosts discussed for IDV have in fact been shown to naturally infect IDV.
